# HilbertA: Hilbert Attention for Image Generation with Diffusion Models

## Abstract

Designing sparse attention for diffusion transformers requires reconciling two-dimensional spatial locality with GPU efficiency, a trade-off that current methods struggle to achieve. Existing approaches enforce two-dimensional spatial locality but often incur uncoalesced memory access. We present HilbertA, a 2D-aware and GPU-efficient sparse attention mechanism. HilbertA reorders image tokens along Hilbert curves to achieve a contiguous memory layout while preserving spatial neighborhoods, and employs a sliding schedule across layers to enable long-range information propagation without repeated or uncoalesced memory access. To further enhance cross-tile communication and positional awareness, HilbertA introduces a small central shared region. Implemented in Triton, HilbertA delivers comparable image quality with significant acceleration over prior methods on Flux.1-dev, demonstrating the feasibility of hardware-aligned two-dimensional sparse attention for high-resolution image generation. HilbertA delivers attention speedups of $2.3\times$ when generating $1024\times1024$ images, and up to $4.17\times$ at $2048\times2048$, while achieving image quality comparable to or surpassing baselines.

## 1 Introduction

Diffusion models (Ho et al., 2020; Song et al., 2020; Dhariwal & Nichol, 2021) achieve state-of-the-art image generation quality. Yet transformer backbones, such as DiT (Peebles & Xie, 2023), introduce quadratic self-attention complexity, leading to substantial inference latency as the number of tokens scales. Sparse attention offers an effective fix by restricting interactions to structured subsets of tokens. Approaches such as sliding windows (Sun et al., 2024) and block-wise patterns (Beltagy et al., 2020) reduce computation and memory while maintaining modeling capacity.

Prior work on sparse attention has largely been shaped by properties in 1D sequence modeling (i.e., text), leaving 2D cases like images underexplored. In vision, an effective sparse pattern should maintain each token's two-dimensional neighborhood (a property we denote as 2D locality), *and* align with GPU memory layouts. This creates a fundamental *dilemma* (Figure 1): memory-contiguous layouts are hardware-efficient but break spatial proximity, whereas 2D-local patterns preserve spatial structure but typically induce uncoalesced memory access. For example, CLEAR (Liu et al., 2024) enforces 2D locality with circular receptive fields, but its indexing produces scattered reads; likewise, Sparge Attention (Zhang et al., 2025) compresses self-similar blocks via a Hilbert curve yet still yields inefficient memory layouts. In both cases, the patterns are 2D-aware but not hardware-aligned, and low-level memory inefficiencies undermine the expected speedups.

To address the dilemma, we introduce **HilbertA**, a hardware-aligned sparse attention that preserves 2D locality with minimal uncoalesced GPU access. HilbertA combines three complementary components as shown in Figure 2. (i) *Reordering*: Tokens are reordered along a Hilbert curve so that spatial neighbors remain adjacent in memory. This choice is quantitatively supported by proposed metrics, showing that Hilbert curves best preserve spatial locality while reducing geometric distortions compared to alternatives. (ii) *Tiling*: the reordered sequence is partitioned into tiles, within which dense attention captures fine-grained local features. This tiling accelerates attention by enforcing sparsity, while the fractal, self-similar nature of Hilbert curves ensures tiles of any size remain spatially coherent (iii) *Sliding*: HilbertA employs a fixed-offset, layer-wise sliding strategy together with a static central shared region to enable structured cross-tile communication. Unlike naive pattern switching, this approach preserves the memory layout and avoids costly reallocations.

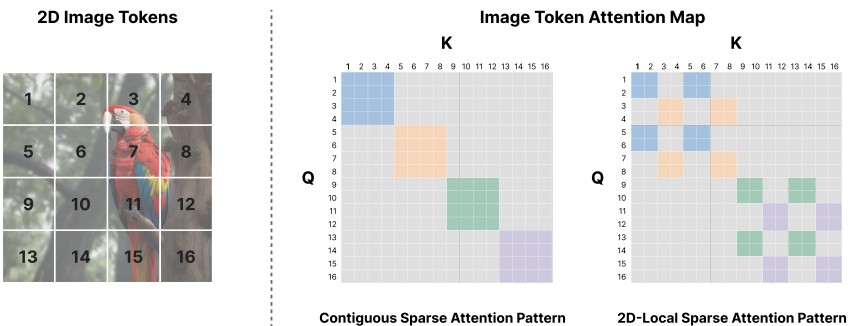

Figure 1: Image token layouts with their corresponding sparse attention patterns. Left: Image token layouts. Sparse patterns (Middle: Contiguous but not 2D-local; Right: 2D-local but not contiguous).

To validate these claims, we evaluate HilbertA on `Flux.1-dev`. Empirical results show that HilbertA achieves up to $2.3\times$ attention and $1.10\times$ end-to-end speedup at $1024 \times 1024$ resolution, and $4.17\times$ and $1.51\times$, respectively, at $2048 \times 2048$. HilbertA achieves these gains even under lower sparsity levels, underscoring the contribution of memory efficiency to the overall acceleration. Meanwhile, HilbertA attains image quality that is both quantitatively and qualitatively comparable to baselines, striking a promising balance between speed and fidelity.

Overall, HilbertA demonstrates that a lightweight, Hilbert-curve–guided layout can concurrently satisfy the three central principles for sparse attention on image: preserving 2D locality, enabling effective cross-tile information exchange, and ensuring coalesced, hardware-efficient memory access—without auxiliary re-indexing structures or costly reallocation.

## 2 RELATED WORK

**Kernel-based and Linearized Attention.** Performer (Choromanski et al., 2021) and Linformer (Wang et al., 2020) introduce kernelization or low-rank factorization to approximate attention with linear complexity. Performer achieves linear complexity through positive orthogonal random features, while Linformer reduces computation by projecting sequences into lower-dimensional spaces. These approaches are complementary to ours.

**Sparse Attention Patterns.** Sparse attention mechanisms reduce computational complexity by limiting token-to-token interactions. Early methods such as Sparse Transformer (Child et al., 2019) and Longformer (Beltagy et al., 2020) adopt fixed windowed or local+global attention patterns, while BigBird (Zaheer et al., 2020) extends these with block-sparse, random, and global connections to enhance model capacity. MInference (Jiang et al., 2024) proposes prefill sparsity by

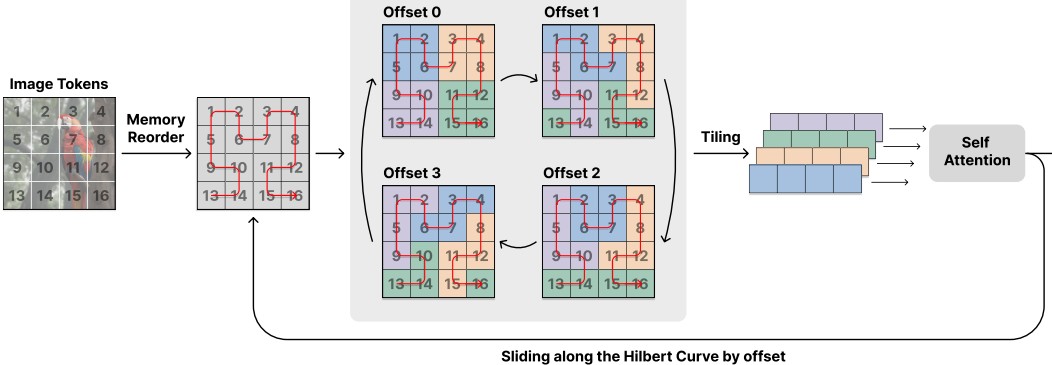

Figure 2: The HilbertA pipeline has three stages: (1) **Reordering** tokens along a Hilbert curve for contiguous memory layout; (2) **Tiling** tokens into local blocks for efficient intra-tile attention; and (3) **Sliding** the window by a fixed offset, enabling cross-tile interaction while preserving efficient memory access.

reducing attention computation to only critical regions. NSA (Yuan et al., 2025) from DeepSeek introduces a trainable sparse mask tailored for prefill acceleration. XAttention (Xu et al., 2025) adaptively selects blocks based on anti-diagonal saliency to balance compute and performance. MoBA (Lu et al., 2025) dynamically merges attention blocks guided by similarity-based objectives, improving routing efficiency across token clusters. However, despite growing interest in sparse attention for language models, few methods have explicitly explored 2D-aware sparse patterns tailored for image generation tasks, where spatial structure is critical.

**I/O-Aware Tiled Attention.** FlashAttention (Dao, 2023) accelerates exact attention by computing in small tiles that fit into fast memory, thereby minimizing memory I/O without sacrificing accuracy. This approach demonstrates how careful memory management can substantially reduce computational overhead. Our method is related in spirit but differs in focus: rather than tiling the attention computation itself, Hilbert Attention enhances efficiency through memory layout optimized by Hilbert-curve token reordering, which preserves spatial locality while ensuring coalesced access. In this sense, our method complements FlashAttention by addressing efficiency from another angle.

**Hilbert Curve.** Hilbert curves are widely used to preserve locality when mapping 2D data to 1D, and have shown benefits for neighborhood structure and coherence in classification and spatial modeling tasks (Erkan & Aksoy, 2023; Tang et al., 2023). SpargeAttention (Zhang et al., 2025) extends Hilbert curves into sparse attention but only as a pre-processing step, with sparsity ultimately dictated by dynamic similarity thresholds; the memory access remains uncoalesced. In contrast, our method directly designs the sparse pattern and propagates information following Hilbert curves. This design preserves contiguous memory access, translating naturally into hardware efficiency.

## 3 METHOD

We propose a lightweight mechanism, *Hilbert Attention*, that simultaneously ensures spatial locality, efficient information propagation, and reduced memory cost. The design integrates three components: Hilbert-curve reordering for contiguous token access, two-dimensional tiling for local attention, and cross-layer sliding to enable structured communication across tiles.

### 3.1 SPACE-FILLING CURVES FOR TOKEN ORDERING

Transformers operate on one-dimensional sequences, whereas images are naturally arranged on two-dimensional grids. This mismatch requires us to "reorder" image tokens into a sequence. Under full attention, the ordering is immaterial because diffusion Transformers are permutation equivariant. But order becomes critical with sparse attention, where each token interacts only within a restricted neighborhood of tokens. In this case, the choice of linearization determines which neighbors fall into the same attention block. Poor ordering can scatter nearby patches across blocks and weaken the model's ability to capture local structure. In contrast, a good ordering preserves the spatial locality of images, making sparse attention more effective.

To make these considerations precise, we model the image grid as a graph $G = (V, E)$, where $V = \{(i, j) \mid 0 \le i < m, \ 0 \le j < n\}$ are the $m \times n$ cells and $E \subseteq V \times V$ connects 4-neighbor adjacencies. Let $N = mn$. A candidate linearization is a bijection $\pi : \{0, \ldots, N-1\} \to V$ that visits each cell exactly once, with inverse $\pi^{-1}(v)$ giving the position of $v$ in the sequence.

The first metric we consider is the *Edge Average Stretch (EAS)*, which measures the average separation in sequence of cells that are immediate neighbors on the grid. Lower values indicate well-preserved spatial locality. Mathematically,

$$EAS(\pi) \ = \ \frac{1}{|E|} \sum_{(u,v) \in E} d_{1\mathrm{D}}(u, v),$$

where the one-dimensional sequence distance is defined as

$$d_{1\mathrm{D}}(u, v) \ = \ \left| \pi^{-1}(u) - \pi^{-1}(v) \right|.$$

Spatial locality alone does not guarantee that the sequence preserves two-dimensional geometry across scales. To capture this, we introduce the *Geometric Distortion Error (GDE)*, which captures

the residual mismatch between sequence distances and true spatial distances after global rescaling. Specifically, we define

$$GDE(\pi) \;=\; \frac{1}{M} \sum_{(u,v) \in S} \Big( \alpha(\pi)\, d_{1\mathrm{D}}(u,v) - d_{2\mathrm{D}}(u,v) \Big)^2,$$

where $M = |S|$, denoting the summation over a chosen set of pairs $S \subseteq V \times V$, and

$$\alpha(\pi) \;=\; \frac{\sum_{(u,v) \in S} d_{1\mathrm{D}}(u,v)\, d_{2\mathrm{D}}(u,v)}{\sum_{(u,v) \in S} d_{1\mathrm{D}}(u,v)^2}, \qquad d_{2\mathrm{D}}(u,v) \;=\; \big\| (i_u, j_u) - (i_v, j_v) \big\|_2,$$

with $u = (i_u, j_u)$ and $v = (i_v, j_v)$ indicating their grid coordinates.

A low GDE indicates that the ordering provides a consistent, globally scaled embedding of the 2D geometry. Low distortion is crucial in diffusion transformers, as maintaining proportional distances allows the attention mechanism to capture both fine-grained textures and broader structural patterns.

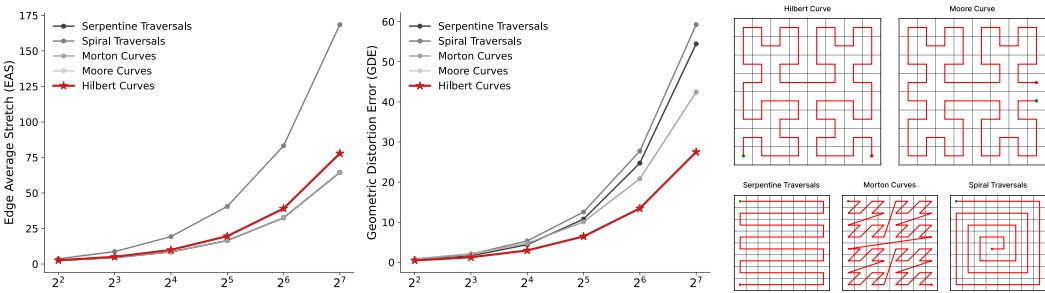

Figure 3: Comparison of space-filling curves. Left & Middle: Edge Average Stretch (EAS) & Geometric Distortion Error (GDE) across grid sizes. Hilbert curves achieve the lowest GDE and the second-lowest EAS, outperforming traversals and Morton order. Right: example orderings at the grid size of $2^3$.

As shown in Figure 3, we evaluate several candidate orderings using the proposed metrics. Serpentine and spiral traversals are simple to construct but introduce long jumps that inflate both edge average stretch and geometric distortion error. Morton (Z-order) curves improve hierarchical coherence, yet tend to fragment local neighborhoods. By contrast, Hilbert curves consistently achieve lower stretch and minimal distortion across grid sizes, owing to their fractal structure. Although Hilbert curves are naturally defined on square grids, we extend our discussion to more general dimensions in Appendix D.

These results provide a principled justification for employing Hilbert curves to reorder image tokens in DiTs. Their ability to preserve spatial locality while reducing geometric distortion makes them especially well-suited to sparse attention, resulting in more accurate and coherent generations.

## 3.2 TILE AND SLIDE

In image generation, sparse attention faces the challenge of jointly satisfying three objectives: (1) preserving fine-grained local details such as edges and textures, (2) enabling information propagation to ensure global coherence, and (3) maintaining computational and memory efficiency. To meet these requirements, we introduce a *tile and slide* mechanism:

**Local tiling for sparse attention.** Given a Hilbert-reordered sequence, we partition it into non-overlapping tiles of $N_T$ tokens and restrict attention within each tile. This strategy preserves the 2D spatial locality of images while bounding computations and memory reads to the tile area rather than the full image, improving efficiency without sacrificing local fidelity. The tile size can be flexibly chosen: smaller tiles enforce stronger locality priors and increase sparsity for higher efficiency, whereas larger tiles provide richer context and better capture longer-range spatial structure. Crucially, because the Hilbert Curve is fractal and self-similar, tiles of any size remain spatially coherent, ensuring they form well-structured neighborhoods in 2D.

**Sliding for information propagation.** Local partitioning alone isolates tokens within tiles, blocking information propagation, and straightforward fixes such as overlapping windows (e.g., CLEAR) or cross-tile attention (e.g., SpargeAttention) typically lead to uncoalesced memory access and reduced efficiency. To address this, we adopt a sliding schedule along the Hilbert-ordered sequence, which preserves contiguous reads while enabling structured cross-tile information exchange.

Formally, let $N$ denote the sequence length after Hilbert reordering, $N_T$ the tile size in tokens, and $T = N/N_T$ the number of tiles. We define tiles as

$$\text{tile}[q] = [\, qN_T,\ (q{+}1)N_T\, ), \quad q \in \{0, \dots, T{-}1\}.$$

Choose a sliding cycle length $L \in \mathbb{N}$ and set the per-layer advance $\Delta = N_T/L$. Each layer advances the attention window by $\Delta$ tokens, so after $L$ layers the window has offset by one tile ($N_T$ tokens), completing a cycle. An example of such a cycle can be found in Figure 2. At layer $\ell \in \{0, 1, 2, \dots\}$, the attention window for a query token at index $i \in \{0, \dots, N{-}1\}$ is

$$\mathcal{A}_i^{(\ell)} = \text{tile}\big[q_i(\ell)\big], \quad \text{where}\ \ q_i(\ell) = \left\lfloor \tfrac{i+\ell\Delta}{N_T} \right\rfloor \ (\text{mod}\ T).$$

As soon as a token from a new tile enters the window, it serves as a messenger: having aggregated the information of its original tile in the previous layer, it carries that information into the new tile. Thus, the effective receptive field (ERF)—the set of tokens that influence token $i$ through a chain of attentions—grows by an entire tile per layer rather than by $\Delta$ tokens. If $q_0 = \lfloor i/N_T \rfloor$ is the initial tile of token $i$, then after $t$ layers,

$$\text{ERF}_i(t) = \bigcup_{k=0}^{\min\{T,\, t\}-1} \text{tile}\big[(q_0 + k)\,(\text{mod}\ T)\big],$$

with size and coverage ratio

$$|\text{ERF}_i(t)| = N_T \cdot \min\{T, t\}, \qquad C_i(t) = \frac{|\text{ERF}_i(t)|}{N} = \min\Big\{1, \tfrac{t}{T}\Big\}.$$

In the example of Figure 4, we set $L = 4$ and $N_T = 4$, so $\Delta = 1$: the window visits four adjacent, two-dimensional neighbor tiles per cycle while maintaining coalesced, contiguous memory access along the Hilbert order. This schedule guarantees that, after $T$ layers, the ERF spans the entire sequence and every token has an indirect path to all other tokens, even with local attention alone.

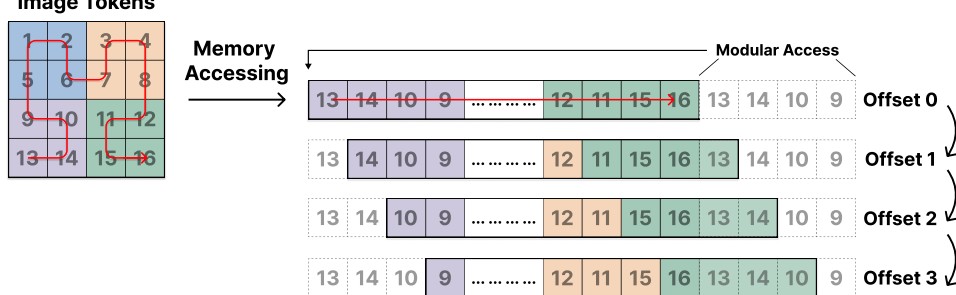

Figure 4: Illustration of sliding tile access along the Hilbert-ordered sequence. Left: image tokens are reordered by the Hilbert curve and partitioned into tiles. Right: modular indexing enables contiguous memory access as the attention window advances, ensuring efficient cross-tile communication without uncoalesced reads.

For maximum efficiency, we use modular indexing: when a read reaches the end of the sequence, it wraps to the head, preserving coalesced access without extra copies. This design reveals an inherent efficiency–fidelity trade-off. The tile size $N_T$ mediates locality versus global context: smaller $N_T$ improves efficiency by limiting computation to finer regions, while a larger $N_T$ enhances global consistency by capturing broader spatial interactions. By tuning $N_T$, one can balance locality, coherence, and efficiency within a fixed memory budget.

## 3.3 SHARED REGION FOR ROPE

To further enhance global coherence, we introduce a fixed shared region located at the image center that every tile attends to. This region serves two complementary roles. First, it facilitates long-range communication by acting as a global relay: each tile can exchange information with the shared region, which in turn aggregates and redistributes context across the entire image. Second, it provides a positional anchor under Rotary Positional Embeddings (RoPE) (Su et al., 2021). While RoPE captures relative positions within each tile, it does not convey where a tile lies in the full image. By designating a shared central region, all tiles gain a consistent reference point, implicitly encoding their location and improving structural consistency in generated images.

This static central region design aligns with our principle of preserving contiguous memory access and avoiding pipeline complexity. A dynamic shared region would necessitate token reindexing and memory copying at every layer, introducing nontrivial computational overhead and disrupting memory contiguity, thereby diminishing the acceleration benefits offered by HilbertA. Also, learned alternatives—such as region predictors or trained shared tokens—introduce additional parameters, training cost, and uncertain generalization across different resolutions. By contrast, a fixed central region is parameter-free, hardware-friendly, and robust across input shapes. It provides a simple yet effective mechanism for information flow and positional grounding without sacrificing efficiency.

## 3.4 TRITON KERNEL DESIGN

With a precomputed Hilbert bijection, token reordering reduces to a single gather operation, performed only twice per inference—once for reordering and once for restoring the original layout. Tile sliding is equally lightweight, implemented as a pointer shift instead of a memory copy.

To fully leverage HilbertA, we implement the sparse attention as a custom Triton kernel. The kernel executes two parallel passes within a single launch: one restricted to local tiles for fine-grained modeling, and another attending to the shared global prefix (text and anchors) for cross-modal context. The shared-region attention is implemented in the style of FlashAttention, where each query block attends to global keys and values. By fusing this computation with the non-shared sparse kernel, we avoid redundant kernel launches and maximize parallelism. This design yields high hardware utilization while preserving the efficiency of our memory layout. Full implementation details and the complete algorithm are provided in Appendix A.

## 4 EXPERIMENTS

### 4.1 EXPERIMENT SETUP

**Setup.** We evaluate HilbertA on the `Flux.1-dev` model using the `Diffusers` framework to generate images at resolutions $1024 \times 1024$ and $2048 \times 2048$. We report three standard quality metrics: LPIPS (Zhang et al., 2018), CLIP-I (Radford et al., 2021), and FID (Heusel et al., 2017). Using 5,000 images generated with a fixed random seed (42) using COCO Val-5k captions (Lin et al., 2014) as prompts, we compute CLIP-I and LPIPS against the original Flux outputs to evaluate semantic and perceptual fidelity, and FID against COCO Test-17k reals to measure distributional alignment. Inference efficiency is measured by the median wall-clock time of the attention module alone and end-to-end generation latency per image on an NVIDIA A100 GPU.

**Training.** We fine-tune `Flux.1-dev` using LoRA with a low-rank configuration of $r = 4$. Training is performed on a curated dataset of 10 k images from the CLEAR training setting, consisting of $1024 \times 1024$ samples generated by FLUX-1.dev. Each image is paired with a self-collected, high-quality textual annotation to better align the model with downstream visual reasoning tasks. For HilbertA, we adopt a configuration of 16 tiles and 4 sliding cycles, which improves cross-tile information propagation while maintaining efficient memory access patterns. LoRA fine-tuning is conducted in two stages: an initial 40 GPU-hours (9 epochs) at $1024 \times 1024$ resolution on 4 NVIDIA H100 GPUs, followed by an additional 40 GPU-hours (3 epochs) at $2048 \times 2048$ resolution to further strengthen high-resolution generation capabilities.[1]

---

[1]Dataset publicly available at: `https://huggingface.co/datasets/jackyhate/text-to-image-2M/resolve/main/data_1024_10K/data_000000.tar`.

| Resolution | Configurations | | Sparsity | Image Quality metrics | | | GFLOPs | Latency | |
| --- | --- | --- | --- | --- | --- | --- | --- | --- | --- |
| | Method | Specs | | FID ↓ | LPIPS ↓ | CLIP-I ↑ | | Attn (ms) | Denoise (s) |
| $1024^2$ | Flux.1-dev | - | 0% | 30.6 | 0.0 | 100.0 | 261 | 1.42 | 13.85 |
| | Sparge Attention | - | 17% | 28.7 | 51.5 | 91.3 | 216 | 2.94 | 14.32 |
| | CLEAR | r= 8 | 95% | 33.0 | 50.0 | 90.0 | 64 | 0.92 | 12.85 |
| | | r=16 | 80% | 32.4 | 47.0 | 91.4 | 81 | 1.20 | 13.31 |
| | | r=32 | 22% | 32.2 | 42.9 | 92.7 | 154 | 1.71 | 14.20 |
| | HilbertA | 4 tiles | 75% | 33.5 | 52.0 | 90.5 | 88 | 0.78 | 12.83 |
| | | 16 tiles | 94% | 31.3 | 56.3 | 87.6 | 49 | 0.62 | 12.57 |
| $2048^2$ | Flux.1-dev | - | 0% | 32.6 | 0.0 | 100.0 | 3,299 | 18.14 | 65.28 |
| | Sparge Attention | - | 14% | 28.2 | 38.3 | 93.0 | 2,830 | 21.44 | 78.64 |
| | CLEAR | r= 8 | 99% | 43.2 | 61.6 | 80.5 | 246 | 4.84 | 47.81 |
| | | r=16 | 95% | 39.3 | 59.3 | 83.4 | 353 | 7.59 | 52.31 |
| | | r=32 | 80% | 36.9 | 59.1 | 83.4 | 724 | 12.55 | 60.65 |
| | HilbertA | 4 tiles | 75% | 38.4 | 51.5 | 84.1 | 1,115 | 7.37 | 50.19 |
| | | 16 tiles | 94% | 47.5 | 57.1 | 78.2 | 496 | 4.36 | 45.19 |

Table 1: Comparison of image quality and efficiency across methods and their specifications at $1024^2$ and $2048^2$. We report sparsity (in percentage), FID, LPIPS, CLIP-I, GFLOPs, and latency of attention in milliseconds and denoising time in seconds. Within each resolution block, the best and 2nd best values for each metric (excluding the Flux.1-dev baseline) are highlighted, respectively. (↑: higher better, ↓: lower better).

To further examine the training capacity and dynamics of HilbertA, we also conduct a train-from-scratch experiment on LightningDiT-B/1 (VAVAE-f16d32) (Yao et al., 2025). The model exhibits stability and convergence comparable to the baseline, achieving competitive image quality. Details of the training dynamics are shown in Appendix F.

**Baseline** As few existing methods target sparse attention for image generation, we compare HilbertA with two representative baselines: CLEAR and Sparge Attention. CLEAR restricts attention to a fixed-radius circular region and aggregates global context through token downsampling. SpargeAttention compresses self-similar $Q, K$ blocks into representative tokens to predict block-level sparsity and applies a softmax-aware filter to skip redundant $PV$ multiplications. It also leverages a Hilbert curve reordering to group spatially adjacent tokens and increase intra-block similarity, but this reordering serves only the block-compression step rather than defining the sparse pattern itself. While both methods incorporate 2D locality, they still suffer from uncoalesced memory access, making them suitable for comparison with HilbertA.

## 4.2 Efficiency and Quality Analysis

**Efficiency Analysis** For efficiency evaluation, we compare Hilbert Attention using different numbers of tiles (4 and 16) and fixed shared region sizes–256 for resolution $1024 \times 1024$ and 1024 for $2048 \times 2048$—against three baselines: Flux.1-dev (FlashAttention-2), CLEAR, and Sparge Attention. For Sparge, we tuned a hyperparameter configuration over 5 prompts, spending 11.5 hours. This configuration involves heterogeneous layer-wise hyperparameters (e.g., per-layer sparsity masks, CDF thresholds), making it impractical to concisely summarize with a single setting.

As shown in Table 1, HilbertA consistently outperforms the baselines in terms of efficiency. At $1024 \times 1024$, HilbertA achieves up to $2.30\times$ acceleration in attention latency and $1.10\times$ in end-to-end runtime. By comparison, CLEAR reaches at most $1.54\times$ / $1.06\times$ (attention / end-to-end, $r = 8$) acceleration, while SpargeAttention is in fact slower than dense attention. At $2048 \times 2048$, HilbertA further improves to $4.17\times$ / $1.51\times$, whereas CLEAR achieves up to $3.75\times$ / $1.48\times$ and SpargeAttention again lags behind. Overall, these results highlight that HilbertA achieves the most substantial efficiency gains, with advantages that become more pronounced at higher resolutions.

**Source of Acceleration** We define sparsity as the fraction of skipped entries in the attention matrix $QK^\top$, i.e., sparsity $= 1 - \mathrm{nnz}_{\mathrm{pat}}(QK^\top)/N^2$, where $\mathrm{nnz}_{\mathrm{pat}}(\cdot)$ counts the number of entries computed under the sparse pattern. In principle, higher sparsity—corresponding to fewer

Table 2: Latency overhead introduced by HilbertA relative to full and sparse attention. For image inputs, the sequence length of image in model is 4096 at $1024 \times 1024$ resolution and 16,384 at $2048 \times 2048$, whereas the text sequence length is consistently 512.

| Seq Len | Reorder (ms) | Recover (ms) | Overhead | | |
|---|---|---|---|---|---|
| | | | Full Attn | Tile4 | Tile16 |
| 4096 | 59.221 | 104.118 | 7.20% | 13.08% | 16.56% |
| 16384 | 264.332 | 290.280 | 1.92% | 4.71% | 7.97% |

computations—should yield greater acceleration. Yet in practice, HilbertA achieves superior speedups even when compared to CLEAR at higher sparsity levels, despite performing more FLOPs and exhibiting lower sparsity in some configurations. This discrepancy reveals that efficiency gains are not governed solely by FLOP reduction or sparsity, but are largely dictated by memory access patterns. HilbertA leverages Hilbert-based ordering to ensure contiguous memory access, whereas CLEAR's overlapping windows and SpargeAttention's runtime regrouping disrupt contiguity, causing uncoalesced reads and additional overhead. Thus, real-world acceleration depends not only on sparsity but also on hardware-friendly layouts, and HilbertA effectively translates sparsity into end-to-end speedups, making it well-suited for practical high-resolution image generation.

**Lightweight Implementation** Also, Hilbert Attention is lightweight. Operations like tile-based sliding are naive memory index shifting and incur minimal cost. The only overhead is the token reordering. To quantify the reordering impact, Table 2 reports the latency of applying and recovering the Hilbert order, as well as its relative overhead compared to attention time. We observe that this cost remains consistently low: it constitutes only 1.9%–7.2% of total full attention time. Importantly, this reordering is applied only twice—once at the input and once at the output, and *independent* of the number of transformer layers or generation steps. In diffusion-based generation, where the model performs dozens of denoising steps (e.g., 28 or more), this one-time cost is amortized across the entire process. In summary, Hilbert-based token reordering introduces a small, bounded overhead that does not scale with model depth or generation length.

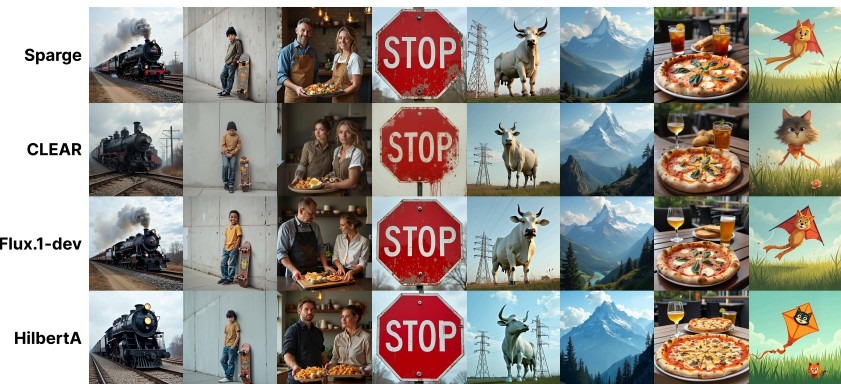

Figure 5: Qualitative results at $1024 \times 1024$ resolution comparing `Flux.1-dev`, CLEAR, Sparge Attention, and HilbertA. HilbertA achieves image quality comparable to state-of-the-art baselines. More qualitative results are shown in the Appendix B.

**Quality Analysis.** At $1024 \times 1024$, HilbertA attains comparable performance to the sparse baselines while delivering the largest speedups. In particular, the 16-tile setting reaches an FID of 31.28—competitive with CLEAR (32.19–33.00) and second only to SpargeAttention (28.67)—with CLIP-I and LPIPS in a similar range to CLEAR. The 4-tile variant is slightly more conservative in quality but remains in the same interval. Importantly, SpargeAttention's quality lead at 1024 comes without efficiency gains, so it does not improve throughput. At $2048 \times 2048$, HilbertA continues to provide quality on par with CLEAR while preserving its acceleration advantage. The 4-tile configuration yields lower LPIPS better than CLEAR and a slightly higher CLIP-I, with FID within the same range (38.41 vs. CLEAR's 36.88–43.25). The 16-tile setting pushes for maximum speed and sacrifices some fidelity, but still remains comparable. By contrast, while Sparge Attention reports

substantially stronger fidelity at 2048, it again fails to produce acceleration; under the efficiency-constrained regime targeted here, such quality improvements are not actionable because they do not translate into better throughput.

We further provide qualitative results in Figure 5, showing samples generated by CLEAR ($r$=8), HilbertA (4 tiles, sliding cycle $L$=4), SpargeAttention, and the original `Flux.1-dev` at $1024 \times 1024$. HilbertA produces images with strong fidelity, on par with the dense baseline, highlighting its ability to preserve high-quality image modeling while delivering efficiency gains.

Across resolutions, HilbertA achieves quality comparable to baseline methods while uniquely delivering consistent and substantial acceleration. In the quality–efficiency trade-off that is critical for practical high-resolution sampling, HilbertA occupies a favorable position, simultaneously offering strong effectiveness and superior efficiency.

## 5 DISCUSSION

### 5.1 EXTENSION TO VIDEO DIFFUSION

The proposed Hilbert curve–based sparse attention mechanism can be naturally generalized to video generation by employing a three-dimensional Hilbert curve that jointly captures spatial and temporal dependencies. Such a 3D Hilbert curve defines a traversal order over the spatiotemporal volume, thereby preserving 2D spatial locality within individual frames while simultaneously maintaining temporal continuity across frames. In this setting, tiles become spatiotemporal blocks, and the sliding attention mechanism extends seamlessly over both space and time. Importantly, the design of the central shared region must also be elevated to the three-dimensional domain under the 3D RoPE. The shared anchors are required not only along the spatial axes (height and width) but also along the temporal dimension to ensure alignment of spatial and temporal references. We regard this extension as a promising direction for future work.

### 5.2 VISUAL ARTIFACTS AT THE BOUNDARY

While HilbertA delivers strong efficiency and image quality, we occasionally observe faint seams aligned with tile boundaries (e.g., subtle vertical or horizontal banding on flat walls or backgrounds), as illustrated in Figure 6. These seams typically lie between two internally coherent regions, track the fixed tile grid, and persist across different sliding offsets, indicating a bias tied to the static partition rather than the per-layer pattern.

A plausible cause is insufficient cross-tile exchange at boundary tokens in rare cases. Although local attention enforces in-tile consistency, cross-tile communication relies on "messenger" tokens and the central shared block; when these tokens aggregate information less effectively from previous tiles, especially at earlier diffusion time-steps, small mismatches can accumulate into visible seams.

This limitation can be mitigated by: (i) distributing the shared region along persistent edges to supply additional cross-tile context; (ii) using interleaved tiling with variable tile sizes or occasional full-attention steps to break boundary regularity and enhance propagation; and (iii) performing more extensive fine-tuning to improve robustness to boundary effects.

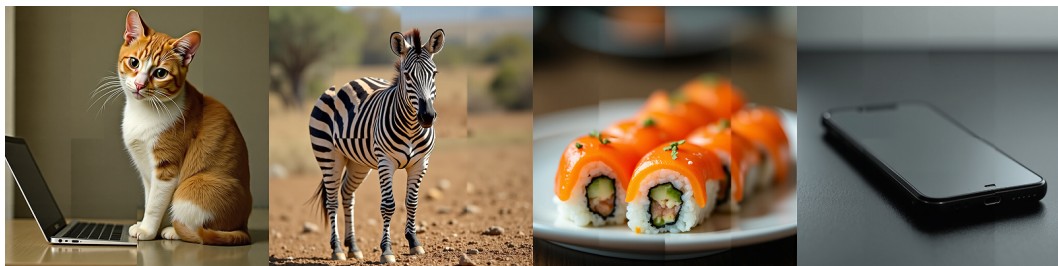

Figure 6: Some cases with visual artifacts at the boundaries of tiles

## 6 CONCLUSION

We present Hilbert Attention, a sparse attention framework that achieves both 2D spatial locality and GPU memory efficiency for high-resolution diffusion models. By leveraging the Hilbert curve to reorder tokens, HilbertA enables coalesced memory access while preserving locality-aware structure. It further introduces an information propagation strategy across layers through sliding along the Hilbert curve, allowing effective context aggregation with minimal overhead. Our experiments show that, with the joint effort of sparsity and memory efficiency, HilbertA significantly reduces attention time and end-to-end latency while maintaining competitive generation quality. Beyond technical contributions, HilbertA lowers the computational barrier for deploying diffusion models and opens opportunities for broader applications in domains like video and medical imaging. At the same time, accelerating generative models necessitates careful consideration of ethical risks such as misinformation and misuse, calling for responsible design and deployment practices.

## 7  THE USE OF LARGE LANGUAGE MODELS (LLMs)

Large Language Models (LLMs) were employed solely as auxiliary tools to polish the writing of this paper. Their role was limited to improving grammar, phrasing, and stylistic clarity. LLMs did not contribute to the conception of research ideas, the design of methodologies, the execution of experiments, or the interpretation of results. All core scientific contributions, including problem formulation, technical development, and empirical validation, were carried out entirely by the authors.

## 8  REPRODUCIBILITY STATEMENT

**Derivation.**  The derivation of the effective receptive field, along with the theoretical justification for adopting the Hilbert curve, is discussed in Section 3.

**Experiments.**  Details of the experimental setup, including the models, datasets, and inference configurations, are provided in Section 4.

**Algorithm.**  The complete implementation of our Triton kernel is presented in Algorithm 1.

**Code.**  The implementation of HilbertA will be made publicly available in the future.

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

## A  FULL ALGORITHM

Our kernel is designed to support `Flux.1-dev`, a DiT-based model that processes both text and image inputs within a unified transformer. To facilitate global information flow and maintain cross-modal alignment, we allow text tokens to attend to all tokens in the same way as we do for the shared anchor region. This identical treatment allows both sets of tokens to be handled uniformly. Therefore, during reordering, the shared-region tokens are gathered and placed together with the text tokens, enabling a single attention pass to process both segments jointly and efficiently.

Within the Triton kernel, we implement sparse attention for non-shared image tokens using two parallel attention passes: one for local-tile attention and one for shared-region attention. In both passes, we begin by loading the same query block. The shared-region pass retrieves key and value tokens from a fixed global prefix of length $N_s$, which includes both the shared anchor region and any textual tokens. This allows image tiles to attend to cross-modal context and global anchors uniformly. In parallel, the local-tile pass loads key and value tokens from a tile-specific segment, with start and end positions determined by a precomputed group ID.

The detailed Algorithm is shown in Alg. 1

---

**Algorithm 1** Sparse Attention Kernel Implementation

---

**Require:** Query $Q$, Key $K$, Value $V$, softmax scale $s$,
Max-logits buffer $M$, Output buffer Out
1: **Configs:** Batch size $Z$, #heads $H$, shared ctx $N_s$, ctx length $N$, head dim $D$,
block sizes $B_M, B_N$, pipeline stage $STAGE$, #groups $G$
2: **for all** block index $b_m \in [0, \lceil N/B_M \rceil)$ **in parallel do**
3:     **(1) Compute per-thread batch and head indices:**

$$\text{flat\_idx} \leftarrow \text{program\_id}(1), \quad z \leftarrow \lfloor \tfrac{\text{flat\_idx}}{H} \rfloor, \quad h \leftarrow \text{flat\_idx} \bmod H$$

4:     **(2) Compute memory offsets for shared vs. non-shared regions:**

$$base\_off \;=\; z \cdot \text{stride\_qz} \;+\; h \cdot \text{stride\_qh},$$

$$off\_shared \;=\; base\_off \;+\; N_s \cdot \text{stride\_qm}, \quad off\_nonshared = base\_off$$

5:     **(3) Compute group size and boundaries:**

$$S \leftarrow \tfrac{N}{G}, \quad \text{group\_id} \leftarrow \left\lfloor \tfrac{b_m \cdot B_M}{S} \right\rfloor,$$

$$\text{start} \leftarrow \text{group\_id} \times S, \quad \text{end} \leftarrow \text{start} + S$$

6:     **assert** $B_M \leq S$ and $S \bmod B_M = 0, \; S \bmod B_N = 0$
7:     Load query block

$$Q_{\text{blk}} = Q[\, z, h, \; b_m \cdot B_M : (b_m + 1) \cdot B_M, \; : \,]$$

8:     Initialize accumulators:

$$m \leftarrow [-\infty]^{B_M}, \quad l \leftarrow [1]^{B_M}, \quad \text{acc} \leftarrow 0^{B_M \times D}$$

9:     **(4) Sparse Attention**

$$K_{\text{shared}} \leftarrow K[z, h, 0 : N_s, :], \quad V_{\text{shared}} \leftarrow V[z, h, 0 : N_s, :],$$

$$start' \leftarrow N_s + \text{start}, \quad end' \leftarrow N_s + \text{end},$$

$$K_{\text{group}} \leftarrow K[z, h, start' : end', :], \quad V_{\text{group}} \leftarrow V[z, h, start' : end', :].$$

10:     Then call

$$(\text{acc}, l, m) \leftarrow \texttt{attn\_inner}(\text{acc}, l, m, Q_{\text{blk}}, K_{\text{shared}}, V_{\text{shared}}, \dots)$$

$$(\text{acc}, l, m) \leftarrow \texttt{attn\_inner}(\text{acc}, l, m, Q_{\text{blk}}, K_{\text{group}}, V_{\text{group}}, \dots)$$

11:     **(6) Finalize and write outputs:**

$$m \leftarrow m + \log_2(l), \quad \text{acc} \leftarrow \text{acc} \,/\, l$$

12:     $M[z, h, b_m] \leftarrow m, \quad \text{Out}[z, h, b_m \cdot B_M : (b_m + 1) \cdot B_M, :] \leftarrow \text{acc}$
13: **end for**

---

## B    MORE QUALITATIVE RESULT

Below in Fig. 7 and Fig. 8, we have shown more qualitative results for HilbertA and different baseline methods across various resolutions.

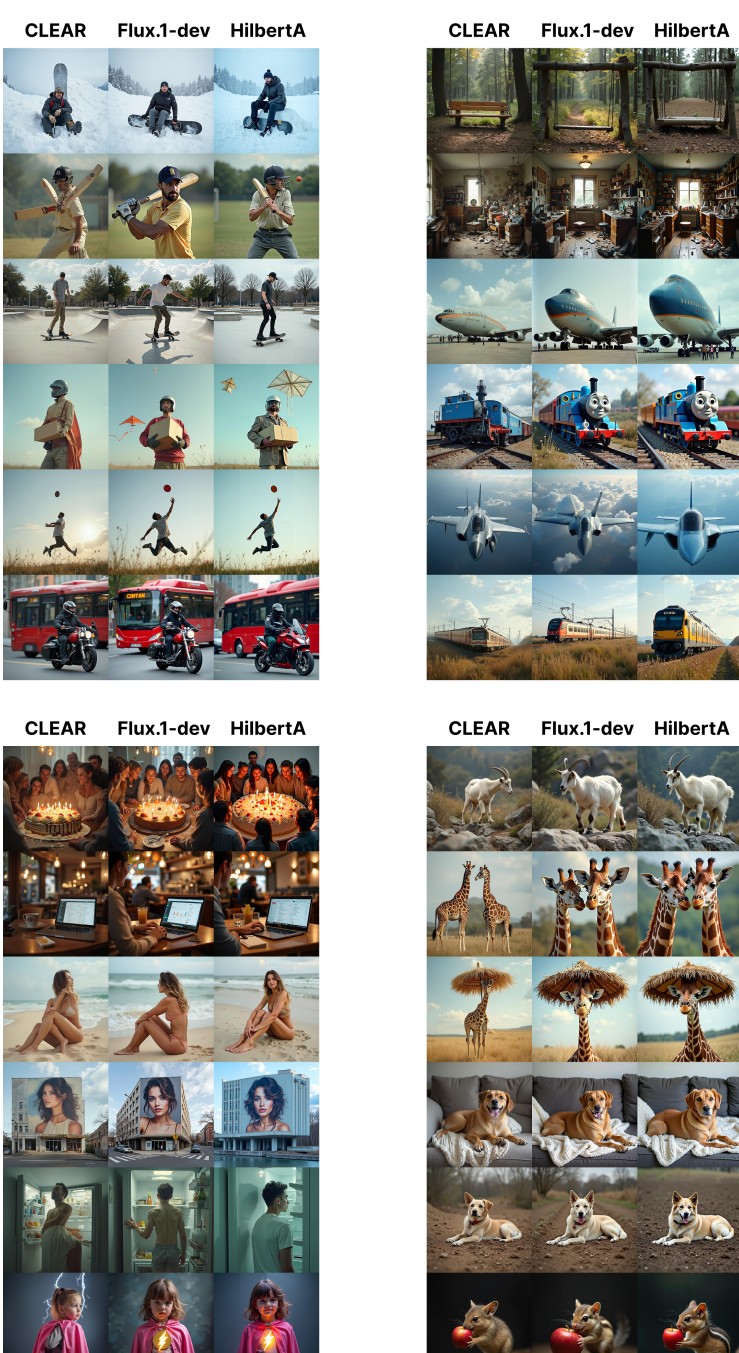

Figure 7: $1024 \times 1024$ Qualitative visual comparison between CLEAR, `Flux.1-dev`, and HilbertA

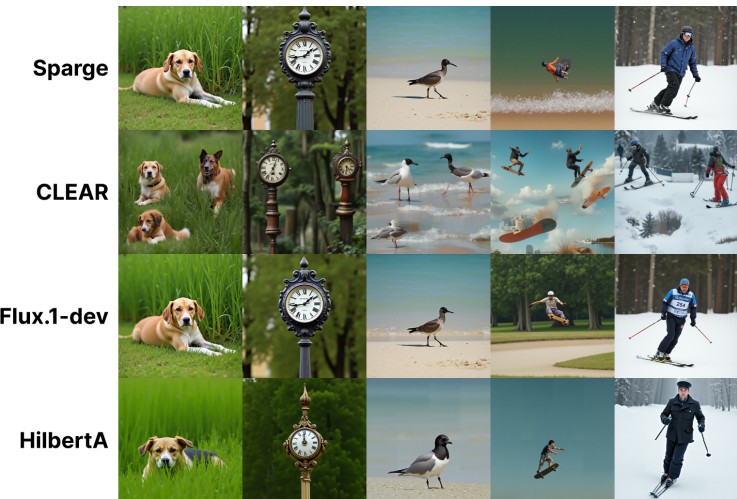

Figure 8: $2048 \times 2048$ Qualitative visual comparison between CLEAR, `Flux.1-dev`, and HilbertA

## C  SHARED REGION VISUALIZATION

In this section, we demonstrate the visualization of our shared region in the image hidden states.

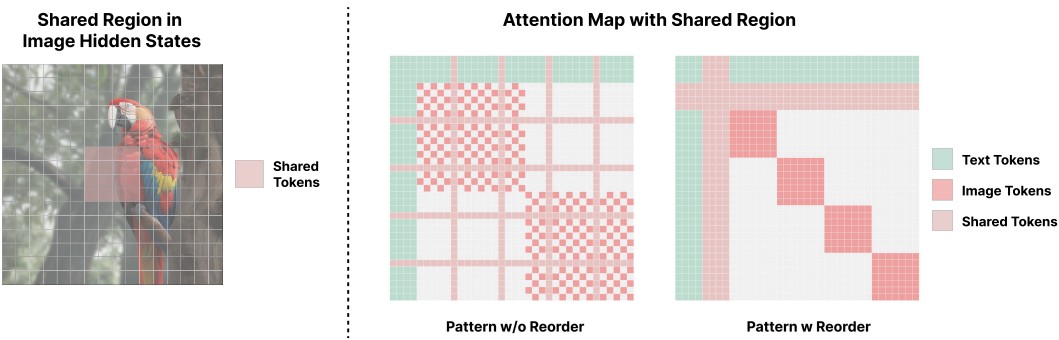

Figure 9: Illustration of attention map with shared region, with and without applying HilbertA reordering.

# D   HILBERT CURVE FOR GENERAL SHAPE

While the standard Hilbert Curve is defined over 2D grids with even (typically power-of-two) dimensions, we note that our method can be naturally extended to handle arbitrary feature map shapes using a simple and GPU-friendly strategy. Specifically, for feature maps with arbitrary spatial dimensions, we construct a Hilbert Curve over the smallest enclosing even-sized grid, and then extract the subregion corresponding to the original shape. The regions outside the original boundaries are treated as padding and masked out during attention computation. This design ensures that all operations are performed on even-dimensioned, regular memory layouts, which are highly compatible with GPU execution patterns and enable coalesced memory access without additional reshaping or copying.

# E  ABLATION STUDIES

To better understand the contributions of individual components in our **HilbertA**, we conduct ablation studies on two key design factors: the *sliding cycle* and the *shared region*. These components are central to HilbertA's information-passing and locality-awareness. We evaluate by generating 300 images using captions from the COCO Val 5k split as prompts. We report the average LPIPS and CLIP-I scores with respect to the original generated images, evaluating perceptual similarity and semantic alignment to assess how well our method preserves visual fidelity and content.

## E.1  SLIDE ANALYSIS

| Resolution | Tiles | Slide Cycle | LPIPS ↓ | CLIP-I ↑ |
|:---:|:---:|:---:|:---:|:---:|
| 1024 | 4 | 2 | 52.1 | 0.904 |
|  |  | 4 | 52.0 | 0.905 |
|  | 16 | 2 | 56.3 | 0.874 |
|  |  | 4 | 56.3 | 0.876 |
| 2048 | 4 | 2 | 51.7 | 0.854 |
|  |  | 4 | 51.5 | 0.841 |
|  | 16 | 2 | 57.4 | 0.785 |
|  |  | 4 | 57.1 | 0.782 |

Table 3: Results for different sliding cycles (2 and 4) under varying resolutions and tile configurations.

We investigate the impact of the sliding cycle by comparing values of 2 and 4 across different resolutions and tile sizes. As shown in Tab. 3, varying the cycle length results in negligible differences in both LPIPS and CLIP-I scores. For instance, at $1024 \times 1024$ with 4 tiles, the metrics remain nearly identical (0.904 vs. 0.905 for CLIP-I; 52.1 vs. 52.0 for LPIPS). These results indicate that our Hilbert-based sliding strategy is largely insensitive to the choice of cycle length, likely due to its strong locality-preserving properties.

We further provide qualitative comparisons of HilbertA with and without the sliding mechanism under the configuration of num_of_tiles = 4 in Fig. 13. Without sliding, the generated images exhibit pronounced boundary artifacts across tiles, resulting in visible inconsistencies. Incorporating sliding substantially mitigates these artifacts by facilitating efficient cross-tile information exchange, thereby producing images with improved coherence and visual consistency. These results underscore the indispensability of the sliding mechanism for achieving high-quality image generation.

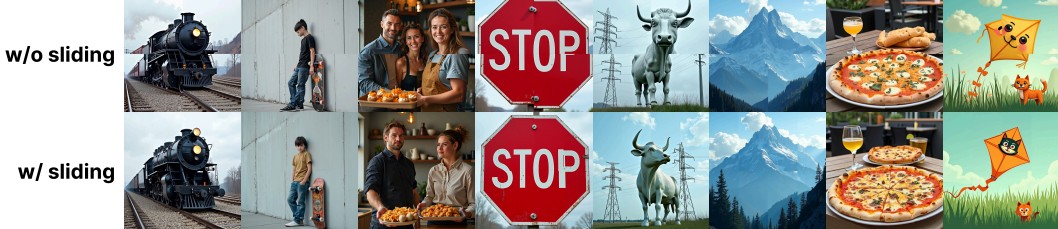

Figure 10: Qualitative comparison of HilbertA with and without the sliding mechanism under the configuration of num_of_tiles = 4 on 8 prompts at $1024 \times 1024$ resolution. **Above:** Without sliding, noticeable boundary artifacts emerge across tiles, leading to visible inconsistencies. **Below:** Incorporating sliding alleviates these artifacts by enabling effective cross-tile information exchange.

## E.2  SHARED REGION ANALYSIS

To test the effect of the shared region on the image, we conduct quantitative and qualitative experiments. From Fig. 11, we find out, through human evaluation, that generations with shared regions

demonstrate better image quality. More specifically, sharing sizes of $16 \times 16$ generates more coherent images compared to the no sharing or sharing with smaller sizes. Also, from Tab. 4 we find out consistent result that $16 \times 16$ in $1024 \times 1024$ and $32 \times 32$ in $2048 \times 2048$ achieves the best score across all setting. Thus, we take these shared region configurations as our default setting.

| Flux.1-dev | No Share | 8×8 Share | 16×16 Share |
|---|---|---|---|

Figure 11: Visual comparison of image generation with different sizes of shared region on $1024 \times 1024$

| Resolution | Shared Region | LPIPS ↓ | CLIP-I ↑ |
|---|---|---|---|
| | 0x0 | 59.2 | 0.871 |
| 1024 | 8x8 | 54.5 | 0.894 |
| | 16x16 | **51.6** | **0.901** |
| | 0x0 | 58.2 | 0.828 |
| 2048 | 16x16 | 54.9 | 0.838 |
| | 32x32 | **51.6** | **0.847** |

Table 4: Results for different shared region sizes under different resolutions.

# F  Training from Scratch

We trained HilbertA from scratch using VA-VAE-f16d32 and Lightning DiT-B/1 on a 100-class subset of ImageNet-1k (140k images), on four A100 GPUs with DDP and a per-device batch size of 512 for 200k steps at a learning rate of $5 \times 10^{-4}$ . The model converged to an FID of 14.10 (with 5k generated images computed against the original distribution), compared to a naive baseline of 13.08. From Fig. 12, both training runs exhibit a rapid initial decline in loss followed by gradual stabilization around 0.30, indicating stable and efficient convergence behavior of HilbertA when training from scratch.

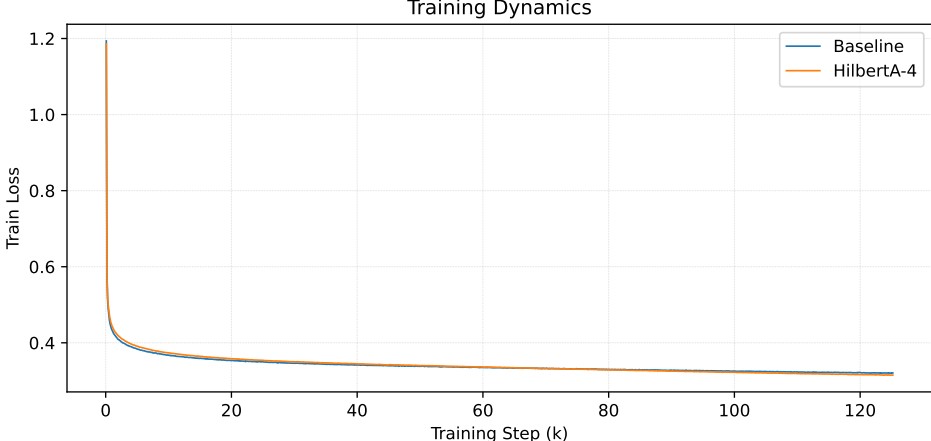

Figure 12: Training loss vs. training step (k) for the baseline and HilbertA-4. Across 0–120k steps, the two curves closely track each other with similar decay and stability, indicating that the Hilbert-curve ordering preserves train-from-scratch trainability comparable to the baseline.

# G    COMPARISON WITH MORE SPARSE ATTENTION METHOD

## G.1    COMPARISON WITH SWIN TRANSFORMER

Although both our sparse attention mechanism and the Swin Transformer restrict attention to local neighborhoods and exchange information through interleaving sparse patterns, their formulations and system-level implications are quite different.

**HilbertA** first permutes the 2D grid into a 1D Hilbert order and then applies attention on contiguous segments along this curve. And sliding along the ordering can yeild up to `tile_size` distinct sparse patterns without any intermediate memory copying, padding, or loss of contiguity. This provides fine-grained control over locality and coverage while preserving a block-aligned dense-matrix layout that is well suited for efficient GPU execution. Moreover, The global reordering is performed only twice—once before and once after generation, no per-layer pattern-dependent memory copies are needed. **Swin**, on the other hand, uses axis-aligned 2D windows and obtains cross-window communication by shifting image diagonally. In practice, this shifting requires copying feature maps into a new memory layout to achieve high efficiency, so the pattern alternation would incur extra memory-copy overhead every time the pattern changes.

We next address whether the Swin rolling strategy can be directly combined with HilbertA. Swin's shifted-window scheme is defined by two distinct 2D tilings of the feature map, a standard tiling and a shifted one, each composed of fixed square windows. The framework of HilbertA, however, is built around a single curve reordering and tiles formed by contiguous 1D segments, which correspond to irregular but locality-preserving 2D regions. As long as we keep this design (one global reorder plus contiguous segments), a simple sliding operation along the curve cannot reproduce the pair of complementary square-window partitions used in Swin. Emulating Swin-style rolling would require either (i) non-contiguous index sets within each tile or (ii) different reordering across layers, both of which break the uniform, memory-contiguous layout that HilbertA explicitly targets. Thus, Swin's rolling cannot be implemented as a straightforward variant of Hilbert sliding without giving up our main efficiency advantages.

To further clarify the relationship to Swin, we implemented a Swin-like baseline by integrating the shifted-window pattern-switching mechanism into the Flux attention module, without using Hilbert reordering or sliding. This baseline mimics Swin's local-window attention and rolling behavior under identical sampling settings. As shown in Table 5,Swin does not match the generation latency or the generation quality achieved by HilbertA.

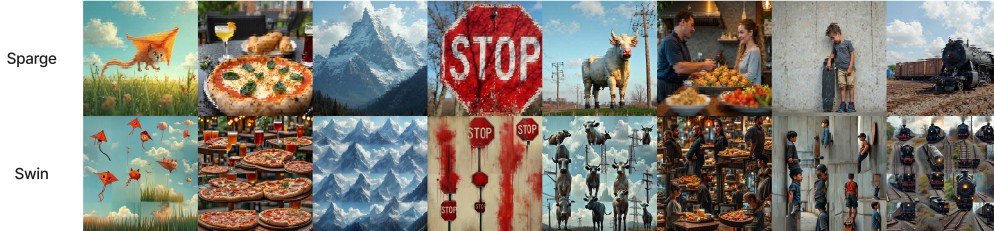

Figure 13: Qualitative comparison between Sparge Attention and Swin-style attention. Sparge retains overall structure but exhibits softened textures and blurred high-frequency details due to similarity-based block merging. Swin-style attention shows window-aligned seams and repeated patterns, reflecting limited cross-window communication.

Figure 13 highlights the corresponding qualitative differences. Swin-style attention tends to produce pronounced block artifacts aligned with its local windows: the pattern-switching mechanism does not propagate information as effectively across neighboring windows, leading to inconsistent global structure, multiple loosely related sub-images, and visible seams between tiles.

Taken together with the quantitative and qualitative results above, these findings demonstrate that Swin-style sparse attention is fundamentally limited in both efficiency and expressive capacity within our setting. Its pattern-switching mechanism incurs unavoidable per-switch memory-copy

overhead and fails to propagate information effectively across windows, leading to suboptimal global consistency and higher FID. In contrast, HilbertA achieves lower latency and better image fidelity through a single memory-contiguous Hilbert ordering and sliding-based pattern diversity, making it a more scalable and robust sparse-attention formulation for large-scale generative modeling.

Table 5: Comparison of Swin-style window attention, HilbertA, and Sparge Attention. Lower FID indicates better generation quality.

| Method | Sparsity (%) | FID ↓ | End-to-End (s) ↓ |
|---|---|---|---|
| Swin (window_size = 16) | 93.75 | 38.52 | 14.62 |
| HilbertA (4 tiles) | 75.00 | 33.50 | 12.83 |
| HilbertA (16 tiles) | 94.00 | **31.30** | **12.57** |
| Sparge Attention | 94.95 | 32.61 | 16.36 |

### G.2 COMPARISON WITH SPARGE ATTENTION

We further tune the Sparge Attention variant to match the sparsity level of HilbertA (94.95% average sparsity) for a fair comparison. As shown in Tab. 5, Sparge Attention achieves a competitive—or even slightly better—FID score. However, its computational characteristics fundamentally limit its practical efficiency in image generation.

Sparge relies on computing block-level similarities and dynamically merging similar regions, which introduces substantial overhead when the sparsity becomes high. While this amortizes well in video diffusion models where sequence lengths are extremely large and attention dominates runtime as reported in their paper, the overhead becomes disproportionately expensive in image generation, where the sequence length is much shorter and attention is not the runtime bottleneck. As a result, the expected sparsity-induced speedup is negated, resulting in slower generation than HilbertA despite having comparable sparsity.

Moreover, although Sparge Attention reports a favorable FID, the generated samples present noticeable visual artifacts due to its similarity-driven block merging. As shown in Fig. 13, the method tends to oversmooth high-frequency regions, leading to blurry textures, washed-out boundaries, and loss of fine detail. For example, the fur of the cat becomes smoothed out, the pizza crust lacks crispness, the mountain ridges lose sharp definition, and the stop sign and cow exhibit diluted surface details. These high-frequency distortions are much more pronounced in images than in videos, further confirming that Sparge's adaptive grouping mechanism does not transfer well to high-resolution image generation.

This comparison highlights the importance of our design philosophy: prioritizing memory-contiguous operations and lightweight reordering enables HilbertA to deliver both high-quality images and real-time generation speed, while avoiding the computational and perceptual drawbacks associated with dynamically computed sparse patterns.

## H MORE RESULTS ON HILBERTA HYBRIDED WITH FULL ATTENTION

In this section, we reported the effectiveness of the hybrid HilbertA Attention with only three steps of Full attention (step: 14, 26, 27) over a total of 28 denoising steps. This step schedule is motivated by our empirical observation that the 11–14 steps are primarily responsible for establishing global image structure, while the final two steps focus on refinement and artifact correction.

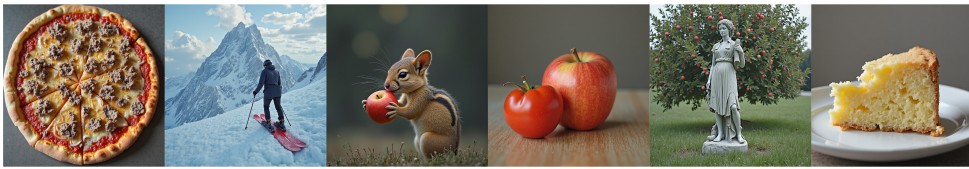

Figure 14: Qualitative result of hybridizing HilbertA with 3 steps of full attention out of a total of 28 steps on 1024×1024 resolution, most artifacts disappeared.

As illustrated in Fig. 14, which shows 1024×1024 samples generated using the hybrid architecture with HilbertA (4 tiles), the resulting images are clean and visually coherent, with noticeably fewer artifacts. This indicates that a small amount of full attention can be effectively incorporated into HilbertA to preserve high image quality.

| Method | FID |
|---|---|
| Flux.1-dev | 30.6 |
| HilbertA (4 tiles) | 33.5 |
| Hybrid | 30.8 |

Table 6: Quantitative comparison of attention strategies. The hybrid method attains near–baseline FID with minor full-attention usage.

Tab. 6 offers supporting quantitative results. Incorporating only a small number of full-attention steps already brings the FID close to the full-attention baseline, showing that HilbertA is an effective backbone and benefits noticeably from even minor full-attention supplementation.

