# OpenReview forum: "Hilbert Attention for Image Generation with Diffusion Models"
_ICLR.cc/2026/Conference — ICLR 2026 Conference Desk Rejected Submission_

### Official Review · Reviewer_k4Tv · 2025-10-30

**Soundness:** 2
**Presentation:** 2
**Contribution:** 2
**Rating:** 4
**Confidence:** 3

**Summary:**

This paper proposes HilbertA, a sparse attention mechanism for image diffusion models that preserves the 2D spatial locality of sparse attention tokens to achieve GPU-efficient implementation by leveraging the Hilbert space-filling curve. The proposed HilbertA reorders patchified tokens along a Hilbert curve to maintain spatial locality and partition them into local tiles. It then applies L sliding window cycles to enable cross-tile communication.

**Strengths:**

- This paper presents a novel application of the Hilbert curve for GPU-efficient sparse attention computation by effectively preserving 2D spatial locality.
- This paper clearly justifies the choice of the Hilbert curve by comparing it against various space-filling curves using two spatial locality metrics: edge average stretch and geometric distortion error.

**Weaknesses:**

- This paper argues that HilbertA can achieve better speedup than CLEAR even with lower sparsity level by preserving 2D spatial locality. Therefore, it is important to evaluate the latency of HilbertA and CLEAR under fair conditions. While the Triton kernel design for HilbertA is described in detail, the latency evaluation setup for CLEAR is not clearly explained. Hence, the fairness of the latency comparison is uncertain.
- The advantage of HilbertA over the conventional CLEAR appears marginal for resolution 1024. In Table 1, for resolution 1024, CLEAR with r=8 and HilbertA with 4 tiles achieve similar end-to-end latencies (12.85s vs. 12.83s), and the image quality metrics are also comparable.
- The main evaluation with fine-tuning is reported only on Flux.1-dev, leaving the generalization capability of the proposed method unclear.
- The term “Seq Len” in Table 2 is used without a clear definition. The experimental setup does not specify how many pixels correspond to a single token, making it difficult to correlate sequence length with image resolution.

**Questions:**

1) Could you provide evaluation results on additional models (e.g., Stable Diffusion 3.5 Large, DiT-XL/2)?

2) How is the latency of CLEAR measured? Please clarify the evaluation setup, including what kind of kernel implementation is used for the latency measurement.

---

> ### Author Response · Authors · 2025-11-26
>
> ***CLEAR Evaluation Setup***
>
> We agree that a fair latency comparison is crucial. For both HilbertA and CLEAR, we benchmark the *attention kernel only* under identical conditions. Specifically, we extract the Q/K/V tensor shapes from a real FLUX.1-dev forward pass and then recreate random tensors of the same shape, dtype, and device for all benchmarks. CLEAR’s official `flex_attn` kernel and our HilbertA Triton kernel are both evaluated on these tensors.
>
> All latency measurements use `torch.utils.benchmark` on the same A100-80GB GPU, in bfloat16, with 1000 forward runs and the median runtime reported. We also sweep CLEAR over its supported window sizes (8, 16, 32) to ensure its strongest configuration is included. Before timing, we clear all unused GPU memory so only the attention kernel is active.
>
> This setup ensures that **HilbertA, CLEAR, and dense SDPA are evaluated on exactly the same tensor shapes and hardware conditions**, making the latency comparison directly comparable. We will describe this evaluation protocol in the appendix for full reproducibility.
>
> ***Generalization to other models***
>
> We appreciate the reviewer’s concern regarding generalization. To show that our findings are not specific to Flux.1-dev, we additionally evaluated our method on a substantially different architecture, LightningDiT-XL. This model differs from Flux.1-dev in both scale (675M vs 12B parameters) and latent resolution ($16\times16$ vs $64\times64$), and therefore represents a strong test of architectural generality. Starting from the public 800-epoch checkpoint, we finetune it for 100k steps on a 100-class ImageNet subset (due to limited time and resources) with a fixed learning rate $1\times10^{-4}$ (batch size 64 per GPU, 4 x RTX6000 Ada, DDP). We then evaluate gFID on 50k generated classifier-free guided samples from the same 100 classes. The results are:
>
> | Methods | FID ↓ |
> | --- | --- |
> | Full Attention | 10.383 |
> | **HilbertA** | **10.442** |
>
> These results confirm that our approach transfers to other transformer-based diffusion models. We will include these results in the revised version. For completeness, we have also trained LightningDiT-B/1 from scratch with HilbertA on the same 100-class subset of ImageNet-1k (140k images). Training was performed on 4× A100 GPUs with DDP and a per-device batch size of 512 for 200k steps at a learning rate of 5×10⁻⁴.
>
> The model achieved an **FID of 14.10** (computed on 5k generated images against the original distribution), compared to a **naive baseline of 13.08**. As shown in **Appendix F Figure 12**, both training runs exhibit a rapid initial decline in loss, and gradual stabilization around 0.30, indicating that HilbertA trains stably and efficiently from scratch.
>
> ***"Seq Len" Misconception***
>
> We thank the reviewer for pointing this out. In the revised version, we have updated **Table 2** and its caption to clearly define the term *“Seq Len”* as the sequence length of the image in the model.
>
> Concretely, for Flux.1-dev model, the downsampling factor is **16x**, so:
>
> - At **1024 × 1024** resolution, it has **Seq Len = 4,096** image tokens.
> - At **2048 × 2048** resolution, it has **Seq Len = 16,384** image tokens.
>
> The **Seq Len** for the prompt is a fixed set of **512** text tokens.
>
> These mappings between image resolution and sequence length are now explicitly stated in the updated Table 2.

---

### Official Review · Reviewer_pCbB · 2025-10-30

**Soundness:** 3
**Presentation:** 3
**Contribution:** 3
**Rating:** 2
**Confidence:** 4

**Summary:**

This paper proposes a new sparse attention mechanism for attention layers in the image generation framework. Specifically, they adopt the Hilbert scan order to reorder the tokens, making the memory continuous and enabling local attention. Besides, they propose to adopt different offsets and a central shared region to enhance and gather cross-tile information. Experiments demonstrate its efficiency.

**Strengths:**

1. adopting hilbert order for sparse attention seems non - trivial.
2. the presentation is clear and easy to follow.

**Weaknesses:**

1. Missing comparison with Swin Transformer. This paper defines a sparse attention mode as each token focusing on interacting with the tokens within a local region. However, this kind of attention is already studied in the Swin Transformer, where each token is restricted to communicate with tokens in the same window. In this view, we highly doubt that using the simple shifted window mechanism would meet all the requirements of the sparse attention in this paper. Each window is local, memory continuous, and could exchange cross - window information using the shifting mechanism. The authors should clearly explain the differences and conduct the experiments to verify the results.

2.  The metric GDE is not reasonable. Since the metric is computed for the whole sequences while the sparse attention only performs attention within a local region (such as the tiling operation in Hiber Attention). This misalignment makes the result in Fig.3 unsolid.

3. Missing quantitative comparison with other scan orders. The author claims "HilbertA reorders image tokens along Hilbert curves to achieve a contiguous memory layout while preserving spatial neighborhoods", However, it seems all the scan orders in Fig.3 can achieve a contiguous memory layout while preserving spatial neighborhoods using tiling and sliding. Missing comparison with those scan orders leads to an unsolid conclusion.

4. Unsolid experiment results. Although the Albert Attention achieves higher efficiency, its results on the image quality lag behind the competitors (especially for Sparse Attention). Why not adjust the hyper - parameter to achieve a better performance and show the latency at that setting? Besides, it is weird that Sparse Attention has a longer latency than Flux.1 - dev since Flux.1 - dev adopts the full attention. It seems that the Sparse Attention implementation is not optimal using Triton code. In such a case, we doubt that the high - efficiency performance of Hilbert Attention comes from the Triton implementation.

We indeed are very interested in this method. However, the above concerns also indeed hinder the acceptance. If the author addresses all the concerns, we would like to raise the score to 6 or higher.

**Questions:**

see above.

---

> ### Author Response · Authors · 2025-11-26
>
> ***Comparison with Swin Transformer***
>
> We thank the reviewer for the insightful comment. Conceptually, both Swin Transformer and our method share a high-level idea of **local attention with a sparse pattern interleaving for image passing**. However, the key difference lies in **memory contiguity**: in HilbertA, the contiguity of tokens is preserved *throughout the entire diffusion process*, while Swin’s window partitioning requires additional **memory-copy operations** to maintain memory contiguity. These copies introduce overhead that prevents Swin’s attention from achieving the same level of efficiency in practice.
>
> We discuss this in detail in **Appendix G.1**, where we analyze the challenge of Swin-like window attention being integrated into the HilbertA framework and report its performance when implemented naively within Flux. The results show that the contiguous-memory design of HilbertA provides a structural efficiency advantage beyond what window shifting alone can offer.
>
>
> ***GDE not reasonable***
>
> While attention is implemented as a local operation, the **model-level** computation is not purely local. Each attention layer only connects tokens within a window, but stacking many layers over many diffusion steps with sliding effectively propagates information along the entire graph induced by the 1D ordering. Thus, what matters is not just whether each tiny neighborhood looks correct in isolation, but how these neighborhoods are ***stitched together globally***. A purely local metric would, for example, rate a serpentine traversal as nearly perfect, because every pixel’s immediate neighbors are preserved, even though the global 2D structure is folded into a snake (top and bottom neighbors far apart).
>
> In contrast, GDE explicitly measures how the 1D distances, after optimal global rescaling, align with true 2D distances across a broad set of pairs. This global view is necessary to reflect the topology of information propagation, and therefore cannot be replaced by a local distortion measure.
>
>
> ***All patterns preserve spatial locality & Missing quantitaive comparison***
>
> We agree that 1D scan orders can be implemented with a contiguous memory layout, but we would like to clarify that our claim is **not about continuity alone**. The key property we target is how well the 1D layout preserves 2D neighborhoods. In this sense, not all scan orders are equivalent. For example, in a serpentine traversal, left-right neighbors are contiguous, but top-bottom neighbors are separated by an entire row. Once local attention is applied, vertical neighbors are **systematically broken** across window boundaries.
>
> By contrast, Hilbert curves preserve both horizontal and vertical adjacency much more isotropically. This difference is exactly what the GDE metric tries to capture:
>
> To better support our argument, we have run an additional experiment on LightningDiT-XL. We take the released 800ep pretrained checkpoint, and finetune it for 100k steps on a 100-class ImageNet subset (due to limited time and resources) with a fixed learning rate $1\times10^{-4}$ (batch size 64 per GPU, 4 x RTX6000 Ada, DDP). We then evaluate gFID on 50k generated classifier-free guided samples from the same 100 classes. The results are:
>
> | Scan Order | gFID ↓ | GDE ↓ |
> | --- | --- | --- |
> | Full Attention | 10.383 | – |
> | **Hilbert Curve** | **10.442** | **2.965** |
> | Morton Curve | 10.501 | 4.747 |
> | Serpentine Curve | 10.552 | 4.397 |
>
> Among the scan orders, Hilbert consistently achieves the best FID, followed by Morton and then serpentine. This ranking mirrors the ordering in Figure 3 given by GDE (Hilbert < Morton < serpentine). The absolute gaps are small here because Lightning-DiT XL operates on a $16\times16$ latent grid $2^4$, where the GDE differences are also modest. As the latent resolution increases, GDE diverges much more sharply across scan orders, so we expect the performance gap to widen correspondingly.
>
>
> ***Potential Misconception of Sparge Attention and Sparse Attention***
>
> First, we clarify that the baseline in **Table 1** is **Sparge Attention**, a specific sparse-attention method, not generic “Sparse Attention.” And our Triton kernel is **only used for HilbertA**, not for Sparge Attention. And from Table 5 in the paper, we can find that the Triton customized attention kernel of HilbertA achieves lower latency compared to other baselines.
>
> Although Sparge Attention can produce good image quality at low sparsity, its **Sparse** **block identification overhead** dominates the computation, which is why it can be slower than full attention in Flux.1-dev. To enable a fair comparison, as discussed in **Section 5.2**, we additionally tuned Sparge Attention to match HilbertA’s sparsity level. With outstanding FID, the results still show visible artifacts and **no actual speedup**, due to Sparge’s intrinsic overhead. This supports our conclusion that HilbertA offers a more favorable efficiency–quality trade-off.

---

> > ### Comment · Reviewer_pCbB · 2025-11-26
> >
> > It appears the revised PDF has not been uploaded, as "Appendix G.1" and "Table 5" are absent from the current version, and "Section 5.2" does not relate to the content regarding "tuning Sparge Attention to match HilbertA’s sparsity level."

---

> > > ### Author Response · Authors · 2025-11-26
> > >
> > > Thank you for bringing this to our attention. It appears the revised PDF was not correctly uploaded in the previous submission. We have now re-uploaded the correct version and double-checked its availability. The new version now includes Appendix G.1, Table 5, and the other sections referenced in our comments. We apologize for the oversight and again appreciate your careful reading.

---

> ### Comment · Reviewer_pCbB · 2025-11-27
>
> Good work. We have no further concerns and decide to raise the score to 6. The reason we cannot raise it further is that we are not experts in attention operations, and other reviewers still have some concerns to be addressed.

---

### Official Review · Reviewer_bhkz · 2025-11-01

**Soundness:** 2
**Presentation:** 3
**Contribution:** 2
**Rating:** 4
**Confidence:** 4

**Summary:**

This paper introduces Hilbert Attention, a sparse attention mechanism designed for diffusion transformers to handle high-resolution image generation. Hilbert Attention rearranges image tokens using Hilbert curves, which helps keep memory access efficient and spatially coherent on GPUs. The method combines local attention tiles, sliding windows, and a central shared region to improve information flow. The author claimed that it speeds up attention computations while keeping the generated image quality acceptable compared to other methods.

**Strengths:**

- The idea of rearranging tokens using Hilbert curves is interesting and helps make GPU memory usage more efficient.
- The method is practical, combining simple components like local tiles and sliding windows in a coherent way.
- Clear experimental results demonstrating real efficiency improvements.
- The analysis of the method (like attention patterns and receptive field) is insightful.
- Applying the method to higher resolutions or multi-image conditions, as suggested, could further highlight its usefulness.

**Weaknesses:**

- Table 1 is crowded and hard to read.
- The generated image quality still noticeably drops compared to dense methods, limiting immediate practical use.
- The experimental settings at 1024×1024 and 2048×2048 resolutions may not fully demonstrate the advantages of Hilbert Attention, larger or more complicated scenarios might show clearer benefits.
- Important training details like LoRA ranks and dataset specifics are missing.
- Testing Hilbert Attention on video generation (mentioned briefly in the paper) could help demonstrate its broader usefulness.

**Questions:**

1. What LoRA ranks did you use in training?
2. Could you give more details on the training dataset?

---

> ### Author Response · Authors · 2025-11-26
>
> ***Table 1 hard to read***
>
> Thank you for the helpful suggestion. We agree that the original Table 1 was dense, so we refined the presentation to improve readability. Specifically, we reorganized the layout and added subtle visual highlighting to indicate the best and second-best entries in each column. This makes the table easier to scan and helps readers quickly identify strong-performing configurations.
>
> ***LoRA and Training Details***
>
> We thank the reviewer for pointing this out. These training details are included in **Section 5.1 (Training Details)** of the paper. Specifically, we use a **LoRA rank of 4**, and all experiments are trained on the **CLEAR self-collected image–text dataset**, for which we also provide a **download link** in the paper. We hope this clarification resolves the reviewer’s concern.
>
> ***Larger scenarios of HilbertA***
>
> We appreciate the reviewer for raising this question. We agree that in scenarios involving larger spatial resolutions or more frames—such as in video diffusion—the efficiency advantage of HilbertA would become even more pronounced, as discussed in the general comment. This is because attention accounts for a significantly higher proportion of computation when the input sequence length increases. However, conducting a full video-diffusion fine-tuning and evaluation requires substantial computational resources, making it difficult to include related experiments within this review cycle.
>
> Regarding higher-resolution image experiments, the current state-of-the-art backbone used in our work, Flux-dev, only supports image generation up to **2048×2048**. This limitation prevents a fair visual-quality comparison at extremely high resolutions. Nevertheless, we profiled our Triton-based HilbertA attention kernel and compared it to FlashAttention at **4096×4096** to illustrate the potential efficiency benefit of HilbertA at larger scales. As shown below, HilbertA achieves a **5.75× speedup** in median runtime:
>
> | **Method** | **Median Runtime (ms)** | **Relative Speedup** |
> | --- | --- | --- |
> | FlashAttention | 747.892 ms | 1.00× |
> | HilbertA (tiles = 4) | 130.100 ms | **5.75×** |
>
> We view this as an important direction for future work. We expect the **efficiency and speed advantages of HilbertA to remain stable** as resolution grows because the Hilbert-curve–based sparse pattern preserves 2D locality and promotes efficient memory access and information flow.

---

### Official Review · Reviewer_Qg5H · 2025-11-01

**Soundness:** 2
**Presentation:** 3
**Contribution:** 2
**Rating:** 4
**Confidence:** 4

**Summary:**

This paper proposes Hilbert Attention, a tailored token purmutation order for local window attention to ensure contiguous memory access when reading tokens within in the same window. To eusure cross-window communication, the paper introduces a sliding mechanism. Experiments demonstrate that the effectiveness of the proposed method on practical acceleration.

**Strengths:**

1. Compared with the original form of window attention, the proposed method provides a smart way to accelerate it by permuting the token order.
2. The analysis of the optimality of the Hilbert order is valid.
3. The writing is generally clear.

**Weaknesses:**

1. The physical meaning of the proposed attention method when $offset\neq0$ is not clear. Although it can bring cross-tile communication, it is not natural. It is like a simple but not elegant extension of $offset=0$ and we have to accept it.
2. The experiment results are not so strong. From the main table, the advantage of the proposed method is not significant in terms of neither performance nor efficiency. And there are many artifacts in the visualized samples. For example, in Fig. 5, the glass behiind the pizza is unnatural, which could originate from the ineffectiveness of the cross-window communication mechanism.
3. The qualitative results of higher-resolution images like 2K are not sufficient.

**Questions:**

Is it possible to combine the proposed permutation order with the rolling strategy in the original SWIN Transformer to devise more effective solutions?

---

> ### Author Response · Authors · 2025-11-26
>
> ***Swin Transformer***
>
> Thank you for the suggestion. We first implemented a Swin-style baseline on the Flux framework to directly compare its behavior with HilbertA. Empirically, the Swin variant exhibits **higher generation latency** and **lower FID scores**. Qualitatively, it also produces **block-shaped artifacts** aligned with its local windows and fails to maintain coherent global structure. These results indicate that Swin’s pattern-switching mechanism does not propagate information effectively across windows, leading to fragmented regional details and visible seams. Additional numerical and visual comparisons are provided in **Appendix G.1.**
>
> The underlying reason for this inefficiency is structural. HilbertA relies on a **single global Hilbert permutation** and **contiguous 1D segments** to preserve memory contiguity and enable efficient sliding-based pattern variation. Swin’s rolling strategy, however, requires **two distinct 2D window partitions**, which cannot be reproduced through simple sliding without introducing non-contiguous indexing or multiple permutations—both of which break the core efficiency properties of HilbertA. Therefore, the two mechanisms are **structurally incompatible**, and a direct combination is not feasible without sacrificing our main advantages. Also, a more detailed discussion can be found in **Appendix G.1.**
>
> ***Offset ≠ 0***
>
> We appreciate the reviewer’s question regarding the physical meaning of using a non-zero offset. We would like to clarify that the offset mechanism is *not* an ad-hoc or inelegant extension of the offset = 0 case.
>
> When offset = 0, each tile attends strictly within its own Hilbert-ordered region, ensuring spatial locality. When offset > 0, the attention window shifts along the Hilbert curve in a deterministic manner. Because adjacent positions on the Hilbert curve correspond to spatially adjacent 2D regions, this sliding schedule introduces **structured, locality-preserving cross-tile communication**, rather than arbitrary long-range jumps.
>
> Importantly, this mechanism preserves:
>
> - **2D spatial coherence:** cross-tile interactions always occur between spatial neighbors;
> - **Memory contiguity:** the sliding window follows the Hilbert ordering, avoiding irregular memory access;
> - **Controlled receptive-field growth:** a single offset parameter expands context in a mathematically predictable way.
>
> Thus, the offset mechanism has a physical interpretation: it implements a *locality-respecting diffusion of information* along the Hilbert trajectory. This design is deliberate—offset ≠ 0 is not a simple extension, but a structured communication schedule that maintains both 2D locality and computational efficiency.

---

### Author Response · Authors · 2025-11-26
**General Comments**

We thank the reviewers for their thoughtful comments. Below, we provide responses to the two major concerns regarding **efficiency** and **generation quality**. Before addressing these points, we would like to emphasize that the goal of this paper is to propose a **new memory-efficient sparse attention design that preserves 2D locality**, offering an alternative solution space for efficient image-generation models.

**(1) On Efficiency.**

Our method accelerates **only the attention module**, while the MLP layers remain unchanged, which is a large portion of the latency in image diffusion. Our method is = fully *orthogonal* to existing approaches such as Token Pruning and Token Merging, and can be combined with them to obtain further acceleration in MLP. Moreover, our experiments are currently limited to image diffusion. When extended to **video diffusion**, where sequence lengths are significantly larger, attention accounts for a higher portion of total computation, and our approach is expected to bring even greater benefits. A systematic exploration of these extensions lies beyond the scope of this paper and is left for future work.

**(2) On Generation Quality.**

Regarding the concern about mild artifacts, we found that interleaving a small number of full-attention steps, specifically at steps 14, 26, and 27 out of the total 28 diffusion steps, effectively mitigates most visual inconsistencies. This design is motivated by our empirical observation that the 11~14 steps are primarily responsible for establishing global image structure, while the final two steps focus on refinement and artifact correction.  And the setting achieves ～1.042**x** acceleration, marginally slower than the full HilbertA ~1.047**x**. This hybrid strategy maintains the advantages of our sparse pattern while supplementing global context where necessary.

We hope these clarifications address the reviewers’ concerns and further highlight the practicality and potential extensibility of our approach.

---

> ### Author Response · Authors · 2025-11-26
>
> To further support our claim regarding the effectiveness of using only a small number of full-attention steps, we have included additional experiments in **Appendix H**. We present both qualitative examples and the FID score for the hybrid configuration with three full-attention steps. The results demonstrate that **artifacts are largely eliminated**, and the **FID closely matches that of the full-attention baseline**. These findings reinforce our conclusion that minor full-attention usage is sufficient to complement HilbertA and achieve high image quality.

---

### Note · Program_Chairs · 2026-01-17
**Submission Desk Rejected by Program Chairs**

The following references in this submission do not refer to real documents and/or have major errors in bibliographic information:

 Kai Sun, Yujun Zhang, Jinpeng Wang, Ziyan Sun, Xun Liu, Lei Zhu, and Dahua Lin. Sta: Scale-aware temporally adaptive sparse attention for efficient image generation. arXiv preprint arXiv:2502.04507, 2024. This paper does not exist.